# Rescue of cognitive function following fractionated brain irradiation in a novel preclinical glioma model

Xi Feng[1,2], Sharon Liu[3,4], David Chen[3], Susanna Rosi[1,2,3,5,6†], Nalin Gupta[3,4†]*

[1]Brain and Spinal Injury Center, University of California San Francisco, San Francisco, United States; [2]Department of Physical Therapy and Rehabilitation Science, University of California San Francisco, San Francisco, United States; [3]Department of Neurological Surgery, University of California San Francisco, San Francisco, United States; [4]Brain Tumor Research Center, University of California San Francisco, San Francisco, United States; [5]Weill Institute for Neuroscience, University of California San Francisco, San Francisco, United States; [6]Kavli Institute of Fundamental Neuroscience, University of California San Francisco, San Francisco, United States

**Abstract** More than half of long-term brain tumor survivors develop irreversible cognitive decline that severely affect their quality of life. However, there is no pre-clinical model that allows long-term assessment of cognition, and there is no treatment which ameliorates cognitive deficits in patients. Here, we report a novel glioma mouse model that offers manageable tumor growth and reliable assessment of cognitive functions in a post-treatment manner. Using this model, we found that fractionated whole-brain irradiation (fWBI), but not tumor growth, results in memory deficits. Transient inhibition of CSF-1R during fWBI prolongs survival of glioma-bearing mice and fully prevents fWBI-induced memory deficits. This result suggests that CSF-1R inhibition during radiotherapy can be explored as an approach to improve both survival and cognitive outcomes in patients who will receive fWBI. Taken together, the current study provides a proof of concept of a powerful tool to study radiation-induced cognitive deficits in glioma-bearing animals.
DOI: https://doi.org/10.7554/eLife.38865.001

*For correspondence:
nalin.gupta@ucsf.edu

†These authors contributed equally to this work

**Competing interests:** The authors declare that no competing interests exist.

## Introduction

Cranial radiotherapy for brain tumor patients is usually fractionated into a series of lower doses in order to reduce radiation-induced normal tissue injury. However, cognitive deficits are still seen in more than half of the patients who received fractionated whole brain irradiation (fWBI) (*Meyers and Brown, 2006*; *Greene-Schloesser and Robbins, 2012*). Despite an overall poor longterm outcome, each year approximately 100,000 brain tumor patients survive long enough to experience radiation-induced cognitive deficits in the US alone (*Greene-Schloesser and Robbins, 2012*). The quality of life of these long-term survivors is severely affected by the impairments in cognitive function. Unfortunately, there is no treatment to ameliorate these adverse effects.

The mechanisms responsible for the loss of cognitive function after fWBI are not well understood. Previous studies have shown that single fraction of WBI induces a number of deleterious effects, including changes in neurogenesis and neuronal structures, activation of microglia and astrocytes, and accumulation of peripheral immune cells in the central nervous system (CNS) (*Monje et al., 2002*; *Mizumatsu et al., 2003*; *Monje and Palmer, 2003*; *Monje et al., 2007*; *Rola et al., 2007*; *Monje, 2008*; *Fike et al., 2009*; *Morganti et al., 2014*; *Feng et al., 2016*). We have previously demonstrated that fWBI, simulating a clinical treatment schedule, recapitulates the findings observed

with a single fraction of radiation. From a mechanistic perspective, there is considerable evidence that the activation of inflammatory pathways is a critical factor in the initiation of radiation-induced brain injury (*Morganti et al., 2014*; *Acharya et al., 2016*; *Feng et al., 2016*; *Moravan et al., 2016*). The Colony-Stimulating Factor 1 (CSF-1) is a cytokine involved in the recruitment/activation of myeloid cell precursors to areas of injury. We have shown that using a CSF-1 receptor (CSF-1R) inhibitor during fWBI resulted in a transient depletion of microglia, a lower number of activated microglia, and lower numbers of inflammatory monocyte accumulating in the CNS. More importantly, use of a CSF-1R inhibitor fully prevented radiation-induced long term cognitive impairments in mice (*Acharya et al., 2016*; *Feng et al., 2016*). However, since activated microglia and peripherally-derived monocytes are often recruited secondary to initiation and growth of brain tumors (gliomas in particular), it is not clear whether radiation-induced cognitive changes are modified or attenuated in tumor-bearing animals (*Pyonteck et al., 2013*; *Stafford et al., 2016*). Demonstrating a rescue of cognitive function following radiation in the setting of brain tumors would have significant translational impact.

Cognitive studies in glioma animal models following radiation treatment have not been previously reported. The primary reason is the technical challenge of producing a glioma-bearing animal that survives sufficiently long enough for cognitive assessment. Low grade glioma models tend to have long, and unpredictable tumor onset with variable growth rates, while high grade glioma models tend to be aggressive with an insufficient time window to permit assays of cognitive function (*Chen et al., 2012*). Most models using human xenografted tumors require an immunodeficient host, which would severely limit the ability to study the role of the immune system in cognitive function. Finally, transgenic mouse models have an inherent inconsistency in tumor development and progression which would preclude the ability to perform cognitive testing at predictable times following radiation treatment. To avoid these limitations, we used a novel high-grade glioma model that allows manageable tumor growth, and feasible assessment of long-term cognitive outcomes. We tested the survival and recognition memory performance 1 month after fWBI and CSF-1R inhibitor treatments. We found that fWBI or CSF-1R treatment alone significantly prolonged the survival of animals after tumor implantation. However, strikingly, memory deficits were detected only in mice received fWBI but not in non-irradiated tumor bearing mice. Finally, we demonstrated that CSF-1R inhibitor treatment was able to prevent fWBI-induced memory loss in glioma bearing animals.

## Results and discussion

### The GL261-DTR glioma model

To create a growth-manageable glioma model we stably transfected Diphtheria Toxin Receptor (DTR) into GL261-luc cell line (*Figure 1—figure supplement 1*). When injected into the mouse brain, the GL261-DTR cells formed tumors indistinguishable from non-DTR expression GL261 cells (data not shown). Six doses of DT injection at 1 µg/mouse/daily significantly prolonged the survival of tumor bearing mice (*Figure 1A*), and resulted in delayed increase of bioluminescence imaging (BLI) signals from tumor cells (*Figure 1B*). Survival was further improved when DT treatment was combined with fWBI (*Figure 1A*). This regimen (DT treatment and fWBI) resulted in a more dramatic decrease in BLI signals from tumor cells; and a new round of DT treatment (1 µg/mouse every other day) slowed tumor growth in four of the five mice with recurrent tumors (*Figure 1C*). These results demonstrate that the GL261-DTR glioma model has a predictable and manageable pattern of tumor growth in vivo, which allows delayed cognitive assessments.

The medium survival of DT treated tumor bearing mice was extended to 31 days (*Figure 1A*). DT treatment was able to temporarily inhibit tumor growth, as evidenced by reduced BLI signals compared to the non-treated group (*Figure 1B*). No side effects beside moderate weight loss (<10%, data not shown) were observed in DT treated mice. In the DT + fWBI group, higher efficacy of tumor growth inhibition was achieved (*Figure 1C*). This paradigm further extended the medium survival to 46 days, more than 4 weeks after the completion of fWBI (*Figure 1A*), thus permitting assessment of cognitive function in these mice at a timepoint that matches our previously reported results using tumor-free models (*Belarbi et al., 2013*; *Feng et al., 2016*).

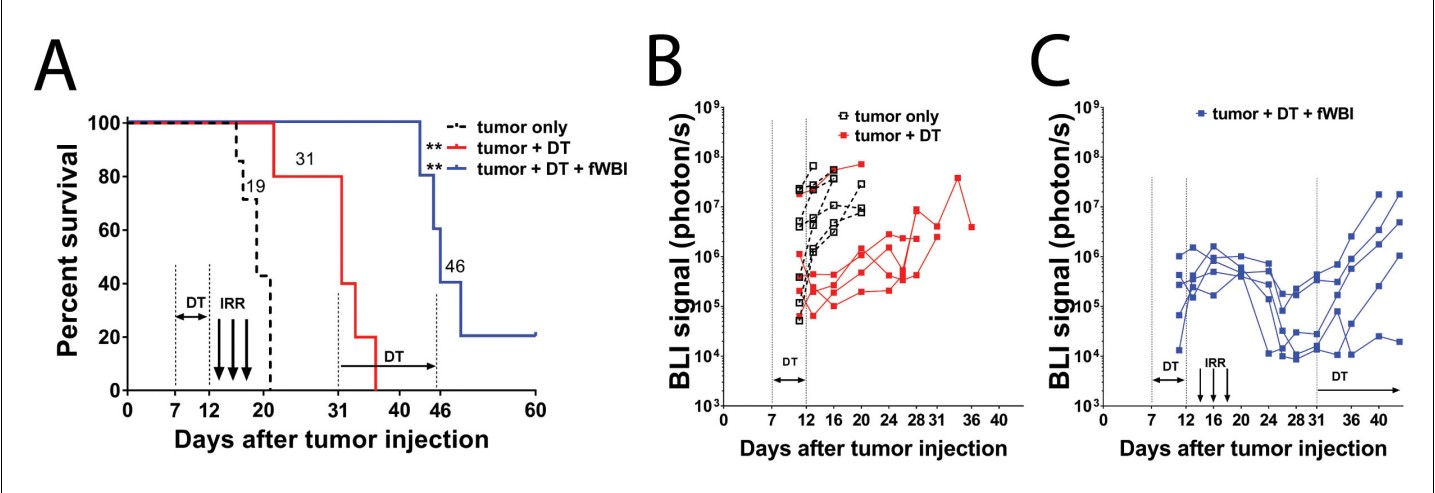

**Figure 1.** GL261-DTR model allows manageable tumor growth. (**A**) Survival curve of mice injected with GL261-DTR cells and the response to DT or combined DT and fWBI treatments. DT treatment was given at 1 ug daily between Day 7 and 12, the combined DT and fWBI treatment group received extra DT treatment at 1 ug every other day between Day 31 and 45. **$p_{tumor + DT}$ = 0.0076, **$p_{tumor + DT + fWBI}$=0.0033, Mantel-Cox Log-rank test compared to the tumor only group. (**B**) BLI of the tumor only and the tumor +DT groups. DT treatment resulted in delayed tumor growth. (**C**) BLI of mice received combined DT and fWBI treatment. Irradiation further delayed tumor growth, and extra DT treatment was able to delay growth of tumors when they recur. (tumor only no treatment control, DT and DT +fWBI treatments were randomly allocated among tumor bearing mice, N = 5 – 7 each treatment group, median survival is labeled next to the curve of each group, sample size was selected to minimize the number of animals used for this pilot experiment.

DOI: https://doi.org/10.7554/eLife.38865.002

The following source data and figure supplement are available for figure 1:

**Source data 1.** source data are included in *Figure 1—source data 1*).
DOI: https://doi.org/10.7554/eLife.38865.004
**Figure supplement 1.** Response of GL261-DTR cells to 48 hr' DT treatment in vitro.
DOI: https://doi.org/10.7554/eLife.38865.003

### Tumor growth in the frontal cortex does not affect recognition memory

We used the Novel Object Recognition (NOR) test to measure recognition memory performance, which partially depends on hippocampal function, in another group of GL261-DTR bearing mice between day 13 and day 16 after tumor implantation. Strikingly, no recognition memory deficits were detected in these mice despite of tumors in their brains (*Figure 2A*). A previous report demonstrated hippocampal dysfunction in mice with subcutaneous sarcoma (*Yang et al., 2014*). In their study, Yang et al demonstrated that there is increased plasma and hippocampal IL-6 concentrations in sarcoma bearing mice compared to tumor-free animals, as well as elevated plasma corticosterone levels. These changes in immune and endocrine systems induced neuroinflammation and were responsible for the observed cognitive deficits. It has been reported that GBM growth can lead to increased plasma IL-6 levels (*Lamano et al., 2017*). However, it is not known whether hippocampal IL-6 levels are also increased with GBM growing in the right frontal cortex. Due to the different strain, age and gender of mice used between the current study and the one reported by Yang et al, it is difficult to make an adequate comparison. It is worth noting that memory deficits are seen in brain tumor patients prior to treatments, but only when brain regions involved in memory functions are directly disrupted by glioma growth (*van Kessel et al., 2017*). Concordantly, in our GL261-DTR glioma model, the tumor growth in the right frontal cortex does not affect recognition memory.

### CSF-1R inhibition during fWBI prevents recognition memory deficits in tumor bearing mice

CSF-1R is expressed in microglia, peripheral monocytes/macrophages and tumor associated macrophages. Pharmacological inhibition of CSF-1R has shown advantages in preventing cranial irradiation-induced cognitive deficits in tumor-free models (*Acharya et al., 2016*; *Feng et al., 2016*), prolonging survival in glioma-bearing mice (*Pyonteck et al., 2013*), and delaying the recurrence of

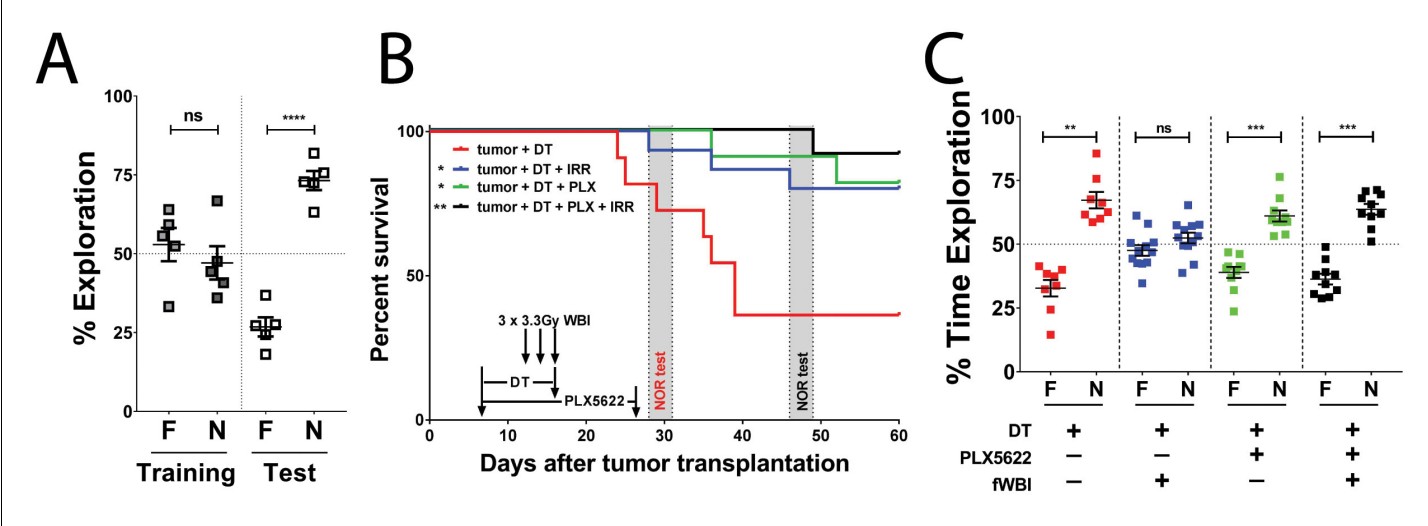

**Figure 2.** CSF-1R inhibitor treatment improves survival and prevents fWBI-induced memory deficits in glioma bearing mice. (**A**) No memory deficits were seen in GL261-DTR bearing mice at 16 days after tumor implantation. ns = not significant, ****p<0.0001, paired t-test. N = 5, error bars show mean ±SEM (**B**) Improved survival of GL261-DTR bearing mice with concurrent DT, fWBI and/or CSF-1R inhibitor treatment. Timeline of treatments and behavior tests are labeled on the X-axis. Novel Object Recognition (NOR) tests were performed between day 28 and 31, or day 46 and 49, for the tumor +DT group or all other groups, respectively. *$p_{tumor + DT + fWBI}$=0.0178, *$p_{tumor + DT + PLX}$ = 0.0167, **$p_{tumor + DT + PLX + fWBI}$ = 0.0029, Mantel-Cox Log-rank test compared to the tumor +DT group, survival curves show combined results from two independent experiments, N = 11–15. (**C**) The results of NOR tests, recognition memory was impaired by fWBI, and CSF-1R inhibitor treatment during fWBI was able to prevent this impairment. The tumor +DT + fWBI group showed memory deficits, this was prevented by CSF-1R inhibitor treatment, combined results from two independent experiments, N = 8–12, ns = not significant, **$p_{tumor + DT}$ = 0.0011, ***$p_{tumor + DT + PLX}$=0.0006, ***$p_{tumor + DT + PLX + fWBI}$ = 0.0001, paired t-test. All treatments were randomly allocated among tumor bearing animals. F = familiar object, N = novel object.

DOI: https://doi.org/10.7554/eLife.38865.005

The following source data and figure supplements are available for figure 2:

**Figure supplement 1—Source data 1.** Source data are uploaded in *Figure 2*-figure supplement-source data 1.

DOI: https://doi.org/10.7554/eLife.38865.007

**Source data 1.** Source data are uploaded in *Figure 2—source data 1*.

DOI: https://doi.org/10.7554/eLife.38865.008

**Figure supplement 1.** (A) DT and fWBI single treatments significantly prolonged survival of tumor bearing mice (**p=0.0021, ****p<0.0001).

DOI: https://doi.org/10.7554/eLife.38865.006

glioma after radiotherapy (*Stafford et al., 2016*). Next, we tested possible treatment interactions using single and combined treatments of DT, CSF-1R inhibitor and fWBI. Treatment with fWBI alone was able to improve survival after tumor implantation, but not to the extent of DT treatment alone, while combined DT and fWBI treatment resulted in better survival than either single treatment (*Figure 2—figure supplement 1A*). CSF-1R inhibitor treatment alone did not prolong survival of tumor bearing mice. With combined CSF-1R inhibitor and fWBI treatments, survival was significantly improved compared to inhibitor alone, but was not significantly different from fWBI alone (*Figure 2—figure supplement 1B*). Therefore, DT, but not CSF-1R inhibitor, acted as a radiosensitizer in our GL261-DTR model, which allowed us to extend survival of tumor bearing mice long enough for cognitive assessment. In light of these results, we tested the effects of CSF-1R inhibition on cognition in GL261-DTR bearing mice after DT and fWBI treatments. We found that DT +fWBI or DT +CSF-1R inhibitor treatment significantly improved survival in the GL261-DTR model (*Figure 2B*, blue and green lines). Combined DT, CSF-1R inhibitor and fWBI treatment did not improve survival time of tumor bearing mice (*Figure 2B*, black line). Most importantly, CSF-1R inhibitor treatment was able to prevent memory deficits induced by fWBI in tumor bearing mice (*Figure 2C*). There was no difference in velocity or time spent in the center of arena (*Figure 2—figure supplement 1C and D*) during the test trials, suggesting that no motor dysfunction or increase of anxiety was caused by tumor growth or treatments at the time of Novel Object Recognition (NOR) test. These results

demonstrate for the first time that tumor-bearing mice are susceptible to fWBI-induced memory deficits, and temporary inhibition of CSF-1R during fWBI is sufficient to prevent radiation-induced deficits. Similar to the tumor-only group, animals in the tumor +DT group also showed no recognition memory deficits (*Figure 2C*), further demonstrating that glioma growth does not affect recognition memory.

## CSF-1R inhibition reduces microglia activation after fWBI

Increased CD11b expression level has been used as an indicator of microglia activation during CNS inflammation (*Hoogland et al., 2015*). To determine the activation status of microglia and their response to CSF-1R inhibitor treatment in the presence of brain tumors, we compared mean fluorescent intensity (MFI) of microglial CD11b using flow cytometry analysis 22–24 days after tumor transplantation (*Figure 3—figure supplement 1*). We found no significant change in microglial CD11b MFI after irradiation compared to non-irradiated mice. Conversely, CSF-1R inhibitor treatment resulted in significant reduction of CD11b MFI compared to untreated groups (*Figure 3A*). Immunofluorescent staining for Iba1, a macrophage specific marker, revealed that no morphological changes of hippocampal microglia were induced by irradiation (*Figure 3D*). However, there was a trend for an increase in the Iba1 stained area in the hippocampus, which was reveresed by CSF-1R inhibitor treatment (*Figure 3B*). Co-staining for microglia activation marker CD68 suggest that radiation induced a significant increase in activated microglia, which was significantly reduced in animals treated with CSF-1R inhibitor (*Figure 3C*). Consistent with the CD11b results, the immunohistochemistry data suggest that treatment with CSF-1R inhibitor reduced microglial reactivity in the hippocampus. Previous reports have demonstrated that cranial irradiation induces morphological changes that correlate with microglia activation, including increased Iba1 +and CD68+cell numbers and staining coverage (*Chiang et al., 1993*; *Acharya et al., 2016*). Taken together with the present results, this work confirmed that cranial irradiation induced microglia activation can be prevented by CSF-1R inhibitor mediated depletion in the presence of brain tumors.

## CSF-1R inhibition alters myeloid cells composition in the contralateral brain of tumor bearing mice

fWBI-induced cognitive deficits has been shown to correlate with increased monocyte accumulation in the CNS (*Morganti et al., 2014*; *Moravan et al., 2016*). We next examined the myeloid composition (namely, resident microglia and peripherally derived macrophages) in the contralateral brains at 6–8 days after the completion of fWBI (see *Figure 3—figure supplement 1* for experimental timeline). fWBI induced significant increase of peripherally derived monocytes in the contralateral brain hemisphere to the tumor (*Figure 4B*). Treatment with the CSF-1R inhibitor induced a partial depletion of microglia, in both irradiated and non-irradiated mice (*Figure 4A*) and prevented radiation-induced infiltration of peripherally derived monocytes (*Figure 4B*). These results are in line with our previous findings in a tumor-free radiation model (*Morganti et al., 2014*; *Feng et al., 2016*). However, in the ipsilateral hemisphere inflammatory monocyte numbers correlate with the size of tumors (*Figure 4—figure supplement 1*). Taken together, these results suggest that changes in contralateral myeloid cells are unaffected by the growth of tumors in the ipsilateral brain hemisphere.

Collectively, we report a novel high-grade glioma model with tumor removal followed by fWBI. The GL261-DTR model allows manageable tumor growth, prolonged survival and feasible assessment of cognition in a post-treatment manner. Using this model, we confirmed findings previously reported in tumor-free models, including increased monocyte accumulation in the CNS and elevated microglia activation. However, strikingly, we found that glioma growth in the frontal cortex does not affect recognition memory. On the contrary, it was the fWBI that induced recognition memory deficits in these tumor-bearing mice. To our knowledge, this report provides the first evidence that CSF-1R inhibition is able to prevent fWBI-induced cognitive deficits in tumor-bearing animals.

It is important to acknowledge that the cell line used for the GL261 model is driven by KRAS and p53, a combination of mutation rarely seen in human glioma patients. These differences in genetic and molecular features may affect the inflammatory microenvironment of the tumor (*Mackay et al., 2018*), and in turn influence cognitive outcomes measured in this study. Further studies using genetically modified mouse glioma models that faithfully recapitulate molecular features in glioma patients would help clarify this concern. In addition, given the existence of Cre-inducible DTR transgenic

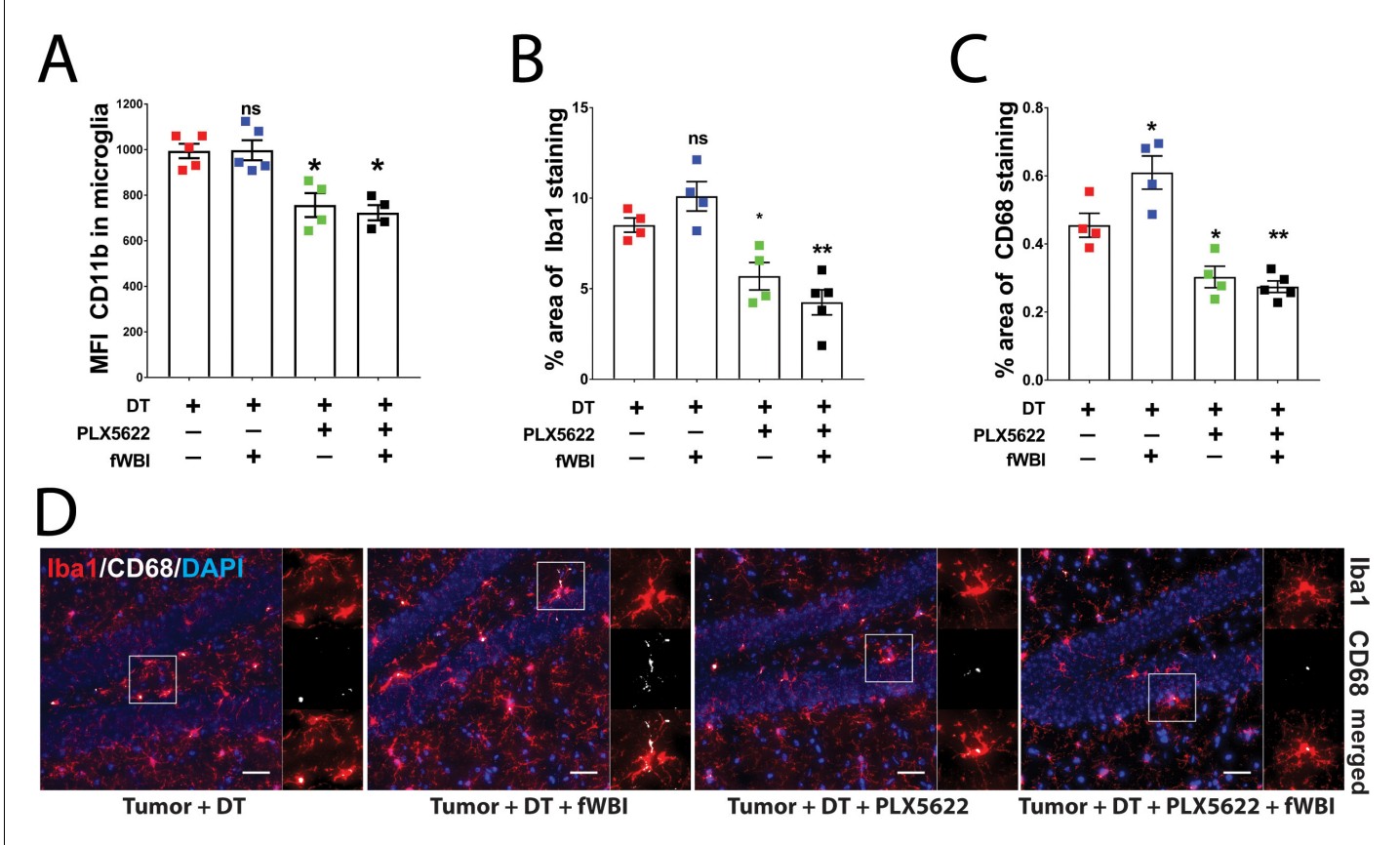

**Figure 3.** CSF-1R inhibitor treatment prevents fWBI-induced microglia activation in glioma bearing mice. (**A**) Plots of the CD11b mean fluorescent intensity of microglia in the contralateral hemisphere of tumor bearing mice. There was significant decrease of microglial CD11b MFI in CSF-1R inhibitor treated mice. there was significant CSF-1R treatment effect $F_{(1,14)} = 38.77$, p<0.0001 with no significant fWBI effect or interaction. (**B**) Quantifications of Iba1 +signal covered area. two-way ANOVA revealed significant PLX5622 treatment effect, $F_{(1,13)} = 39.14$, p<0.0001, no significant radiation effect $F_{(1,13)} = 0.01122$, p=0.9173 and significant interaction between PLX5622 treatment and fWBI, $F_{(1,13)} = 4.793$, p=0.0474. There was a trend of increase in the tumor +DT + fWBI group (p=0.3044), and significant reduction in the tumor +DT + PLX group (*p=0.0381) and the tumor +DT + PLX+fWBI group (**p=0.0017), compared to the tumor +DT group. (**C**) Quantifications of CD68 +signal covered area. Two-way ANOVA revealed significant CSF-1R inhibitor treatment effect, $F_{(1,13)} = 53.07$, p<0.0001, no significant radiation effect $F_{(1,13)} = 3.559$, p=0.0817, and significant interaction $F_{(1,13)} = 7.529$, p=0.0167. (**D**) Representative images of Iba1 and CD68 staining, showing Iba1 staining in red, CD68 staining in white and DAPI in blue. Scale bar = 50 um. (A-C, Two-way ANOVA with Dunnett's multiple comparisons test vs the tumor +DT group, ns = not significant, *p<0.05, **p<0.01. each dot represents value from one mouse (**A**) or the mean values of Iba1 and CD68 (**B and C**) staining quantification of 3 snapshots in the hippocampus, N = 4–5. Error bars show mean ±SEM.

DOI: https://doi.org/10.7554/eLife.38865.009

The following source data and figure supplements are available for figure 3:

**Figure supplement 1—Source data 1.** Source data are uploaded in *Figure 3*-figure supplement-source data 1.
DOI: https://doi.org/10.7554/eLife.38865.011
**Source data 1.** Source data are uploaded in *Figure 3—source data 1*.
DOI: https://doi.org/10.7554/eLife.38865.012
**Figure supplement 1.** Tumor growth after treatment and timeline of flow cytometry analyses.
DOI: https://doi.org/10.7554/eLife.38865.010

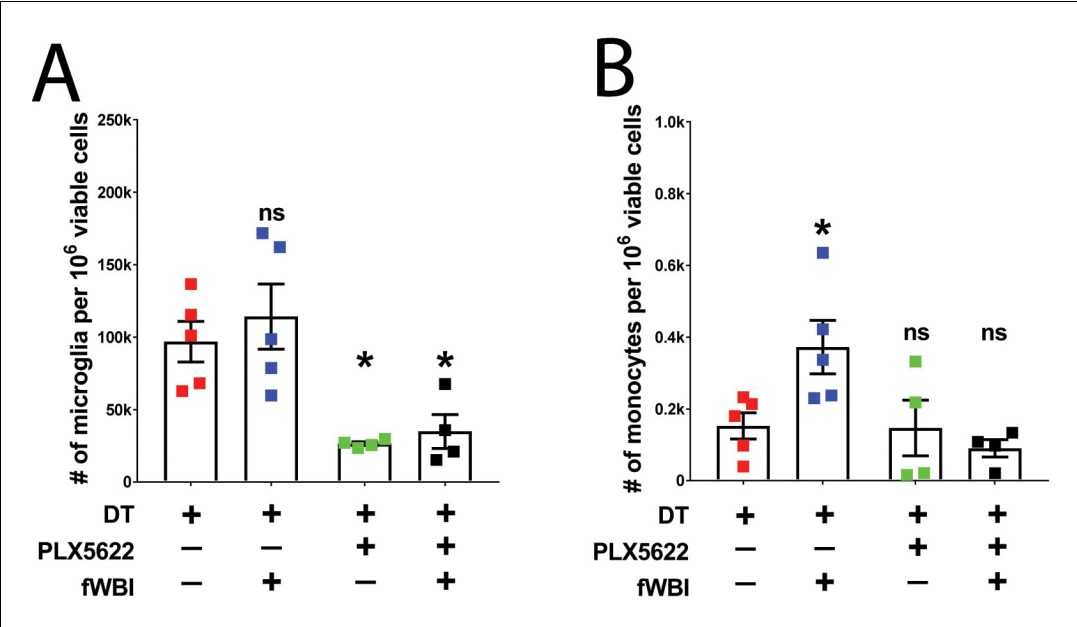

**Figure 4.** Treatment with a CSF-1R inhibitor alters myeloid cell composition in the contralateral hemisphere. (**A**) Plot of microglia (CD11b + CD45 lo) numbers in treated tumor bearing mice. There was significant CSF-1R inhibitor treatment effect ($F_{(1,8)} = 31.45$, $p=0.0005$), but no significant radiation effect or interaction between these two treatments. Two-way ANOVA with Dunnett post-hoc comparisons against the tumor +DT group, ns = not significant, *$p<0.05$. (**B**) Plot of inflammatory monocyte (CD11b + CD45hiLy6Chi) numbers in treated tumor bearing mice. There was significant radiation effect ($F_{(1,8)} = 10.68$, $p=0.0114$) and CSF-1R inhibitor treatment effect ($F_{(1,8)} = 23.49$, $p=0.0013$), as well as significant interaction ($F_{(1,8)} = 5.52$, $p=0.0467$) between these two treatments, two-way ANOVA with Dunnett post hoc comparisons against the tumor +DT group, ns = not significant, *$p<0.05$. Error bars show mean ±SEM values. Each dot represents cell count from a mouse, N = 4–5.
DOI: https://doi.org/10.7554/eLife.38865.013

The following source data and figure supplements are available for figure 4:

**Figure supplement 1—Source data 1.** Source data are uploaded in *Figure 4*-figure supplement-source data 1.
DOI: https://doi.org/10.7554/eLife.38865.015

**Source data 1.** Source data are uploaded in *Figure 4—source data 1*.
DOI: https://doi.org/10.7554/eLife.38865.016

**Figure supplement 1.** Changes in myeloid cells in the ipsilateral hemisphere and growth of tumors with single treatments.
DOI: https://doi.org/10.7554/eLife.38865.014

mouse strains and the fact that human cells intrinsically express DTR, our model could be adapted to other primary mouse brain tumor models as well as xenograft models. Therefore, the current study provides a proof of concept of a powerful tool to study radiation-induced cognitive deficits in glioma-bearing animals.

# Materials and methods

**Key resources table**

| Reagent type (species) or resource | Designation | Source or reference | Identifiers | Additional information |
|---|---|---|---|---|
| Strain, strain background (Mus musculus), C57BL/6J | C57BL/6J | Jackson Laboratory, cat# 000664 | cat # 000664 | |

*Continued on next page*

*Continued*

| Reagent type (species) or resource | Designation | Source or reference | Identifiers | Additional information |
|---|---|---|---|---|
| Cell line (Mus musculus), C57BL/6 | GL261 | NCI-Frederick DCTD tumor/cell line repository | Glioma 261 | Mycoplasma negative |
| Transfected construct | ATF5-DTR-GFP | PMID: 20495567 | NA | |
| Antibody | AF700-CD11b | BD Biosciences | cat# 557960 | Flow Cyt: 1:100 |
| Antibody | BV711-CD45 | BD Biosciences | cat# 563709 | Flow Cyt: 1:100 |
| Antibody | CD68 | Bio-Rad, | cat#: MCA1957 | IF: 1:200 |
| Antibody | Iba1 | Wako chemicals, | cat# 019–19741 | IF: 1:100 |
| Commercial assay or kit | Neural Tissue Dissociation kit (P) | Miltenyi Biotech | cat# 130-092-628 | |
| Chemical compound, drug | CSF-1R inhibitor | Plexxikon Inc | PLX5622 | 300ppm in AIN-76A standard chow |
| Other | AIN-76A standard chow | Research Diets Inc | cat#: D10001i | |
| Chemical compound, drug | Percoll | Sigma | cat# P1644-100ML | |
| Chemical compound, drug | Diphtheria toxin | Sigma | cat# D0564-1MG | |
| Software | Ethovision XT | Noldus Information Technology | Version 11.5 | |

## Mice

All animal experiments were conducted in compliance with protocols approved by the Institutional Animal Care and Use Committee at the University of California, San Francisco (UCSF), following the National Institutes of Health Guidelines for animal care. 11 weeks-old C57BL/6J male mice were purchased from the Jackson Laboratory, and housed in UCSF animal facility for one week before tumor transplantation.

## Treatments

### CSF-1R inhibitor treatment

Control and PLX5622 (300 ppm formulated in AIN-76A standard chow, Research Diets, Inc.) chows were provided by Plexxikon Inc (Berkeley, CA). Approximately 1.2 mg of PLX5622 was ingested by each mouse per day (calculation based on 4 g/mouse chow daily).

### Diphtheria toxin treatment

DT (Sigma-Aldrich) was dissolved in sterile saline at 10 µg/ml. 6 injections of 100 µl DT solution was given to each mouse by intraperitoneal injection every day between day 7 and day 12 (*Figure 1*), or every two days between day 7 and day 17 (*Figures 2–4*) after tumor implantation.

### Radiation treatment

mice were anesthetized and received 10 Gy head-only irradiation in three fractions from a cesium-137 source as previously described (*Feng et al., 2016*).

## Stable transfection of GL261-DTR cells and intracranial transplantation

The GL261 cells were obtained from the NCI-Frederick DCTD tumor/cell line repository. Firefly luciferase reporter modification for bioluminescent imaging was done as previously described (*Clark et al., 2016*). Two micrograms of lineazed ATF5-DTR-GFP plasmid (*Sheng et al., 2010*) was transfected into GL261-luc cells in a 6-well plate using lipofectamine 2000 (Invitrogen) following manufacturer's instructions. Medium supplemented with 200 ug/ml Zeocin (Gibco) was changed every two days to select positively transfected cells. GFP expressing GL261-DTR cells were sorted using a BD Aria III and washed with serum-free RPMI medium before injection. 250,000 cells were injected into the right frontal cortex at the coordinate Bregma, AP + 2 mm, ML +2 mm and DV −2 mm. DT treatment started at 7 days after tumor injection, at 1 μg/mouse/daily for 6 days (*Figure 1*) or at 1 μg/mouse every other day for six doses (*Figures 2*, *3* and *4*).

## Novel Object Recognition (NOR) test

NOR test was performed during the dark cycle as previously described (*Feng et al., 2016*). Briefly, trials from the four-day test were recorded by an overhead camera and analyzed by an automatic video tracking system (EthoVision, Noldus). Animals with insufficient exploration (less than total of 3 s interacting with both objects) during training (day 3) or test (day 4) were excluded. Same object sets were used in NOR tests conducted at 13-16 days or 4 weeks tumor implantation (*Figure 2A* and *Figure 2C*, respectively), or 4 weeks after the completion of fWBI *Figure 2C*. Sample sizes (N > 8) were chosen according to our previous experiments where we can detect radiation effects on memory performance.

## Flow cytometry analysis

Mice were euthanized and perfused with ice-cold PBS. Brains were quickly removed and half hemispheres were placed in ice-cold HBSS. Brain samples were dissociated using the Neural Tissue Dissociation kit (P) according to the manufacturer's instructions (Miltenyi Biotech). Dissociated cells were then resuspened in 30% Percoll (Sigma) solution diluted with RPMI medium (Gibco) and centrifuged at 800 g for 30 min at 4° C. Cell pellets were washed with RPMI medium and resuspended in FACS buffer (1 x DPBS with 2% FBS and 0.5% BSA fraction V) and blocked with mouse BD Fc block (5 μg/ml, purified rat anti-mouse CD16/CD32, BD Pharmingen) for 30 min on ice. Cells were then stained with fluorophore-conjugated antibodies (BV711-CD45, AF700-CD11b). Data were collected on an Aria III sorter (BD) and analyzed with Flowjo v10 (Tree Star Inc.).

## Immunohistochemistry staining and imaging

After perfusion, the other brain hemispheres were immediately put into 4% PFA at 4° C, fixed overnight and then switched into 30% sucrose solution for at least two days. Samples were sliced into 20 μm coronal sections and stained with rabbit anti-Iba1 (1:100, Wako Pure Chemicals) and rat anti-CD68 (1:200, BioRad) followed by incubation with AF-568 goat anti-rabbit and AF-647 goat anti-rat secondary antibodies (1:500, Life Technologies). DAPI was used for nuclear counterstaining. Hippocampal images were taken using a Zeiss Imager Z1 under a 20x objective lens. The area of Iba1 +and CD68+signals were measured in ImageJ.

## Acknowledgements

We thank Dr. Zhi Sheng from Virginia Tech University for providing the ATF5-DTR-GFP plasmid.

## Additional information

### Funding

| Funder | Grant reference number | Author |
|---|---|---|
| National Cancer Institute | R01CA213441 | Susanna Rosi |
| Pediatric Brain Tumor Foundation | PBTC_InstituteAward | Nalin Gupta |

The funders had no role in study design, data collection and interpretation, or the decision to submit the work for publication.

### Author contributions
Xi Feng, Conceptualization, Data curation, Formal analysis, Investigation, Visualization, Methodology, Writing—original draft, Writing—review and editing; Sharon Liu, Investigation, Methodology; David Chen, Data curation, Formal analysis; Susanna Rosi, Nalin Gupta, Conceptualization, Supervision, Funding acquisition, Writing—original draft, Writing—review and editing

### Author ORCIDs
Xi Feng http://orcid.org/0000-0002-6920-1519
Susanna Rosi https://orcid.org/0000-0002-9269-3638
Nalin Gupta http://orcid.org/0000-0001-9539-7052

### Ethics
Animal experimentation: This study was performed in strict accordance with the recommendations in the Guide for the Care and Use of Laboratory Animals of the National Institutes of Health. All of the animals were handled according to approved institutional animal care and use committee (IACUC) protocols (AN145279 and AN 142109) of the University of California, San Francisco. All surgery was performed under Ketamine/Xylazine anesthesia, and every effort was made to minimize suffering.

### Decision letter and Author response
Decision letter https://doi.org/10.7554/eLife.38865.019
Author response https://doi.org/10.7554/eLife.38865.020

## Additional files

### Supplementary files
• Transparent reporting form
DOI: https://doi.org/10.7554/eLife.38865.017

All data generated or analysed during this study are included in the manuscript and supporting files. Source data files have been provided for all Figures.

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
