## [Decision Letter]

Thank you for submitting your article "Rescue of cognitive function following fractionated brain irradiation in a novel preclinical glioma model" for consideration by *eLife*. Your article has been reviewed by two peer reviewers, and the evaluation has been overseen by a Reviewing Editor and Sean Morrison as the Senior Editor. The reviewers have opted to remain anonymous.

The reviewers have discussed the reviews with one another and the Reviewing Editor has drafted this decision to help you prepare a revised submission.

Summary:

The authors aim to investigate cognition in a glioma animal model following fractionated brain irradiation, a paradigm they claim previously difficult to assess due to variability in glioma burden and survival outcome. The authors find that microglial depletion during and after radiation ameliorates cognitive deficits associated with radiation of frontal lobe gliomas.

Studying the cognitive effects of radiation therapy in the context of a glioma is a needed advance in the field. Based on data generated from this model the authors claim that the CSF-1R inhibitor PLX5622 (PLX) enhances the radio response of GL261 orthotopic xenografts and protects against radiation-induced impairment of recognition memory. This study highlights a possible therapeutic strategy for addressing the known effects of microglial inflammation on cognitive function after cranial radiation. While this new model is a useful tool to investigate the contributions of tumor burden to radiation-induced cognitive decline, there were several questions and concerns raised that would need to be addressed before publication, as summarized below.

Essential revisions:

1) This DT- Glioma model is useful as it allows for long term evaluation of cognitive deficits and provides proof of principle; however there are several caveats of the GL261 cell line raising questions regarding clinical relevance. Validating key findings using another documented primary tumor model would strengthen the study and claims, however If this is not feasible, it is important to clearly state the caveats and address points raised by reviewer 2 (below) in the Discussion.

2) The reduction in radiation-induced memory loss is apparent; however data regarding tumor radio response are incomplete and detracts from the strength of this study.

Are the effects of radiation alone and PLX alone independent or additive in prolonging survival in tumor bearing mice? Please address specific comments raised by reviewer 3 (below) related to Figure 2B and provide the necessary data and analyses (radiation only vs. radiation + PLX; PLZX only vs. radiation + PLX).

3) More in depth and quantitative analyses of microglia numbers using resident and peripheral markers in each of the four conditions in Figure 3. It is difficult to assess from images shown in Figure 3B as mean fluorescent intensity if CD11b is not quantitative or a robust indicator of microglia 'activation'.

Reviewer #2:

- While the DT-mediated glioma growth control paradigm is novel, simply using a less aggressive – and more molecularly faithful – model of glioma would similarly allow for long term evaluation of cognition after radiation in the context of microglial depletion. My chief concern is the use of GL261 cells – a murine cell line from the 1990s driven by a KRAS mutation (a rare mutation in human gliomas). The molecular characteristics of the tumor can dramatically influence the inflammatory microenvironment of the glioma (see the recent report from the HERBY study, Mackay et al., 2018), and thus the potential influences of the tumor on cognition. Many well-documented xenograft and primary tumor mouse models exist that allow for assessment of cognition at 1 month post-radiation treatment and could be included in this paper to strengthen the claim that microglial depletion can rescue hippocampal-dependent (or hippocampal-independent) cognition following brain irradiation in the context of molecularly definable, faithful glioma models.

Reviewer #3:

- Whereas the reduction in radiation-induced memory loss is apparent, data regarding tumor radio response are incomplete and detracts from the strength of this study.

The model uses diphtheria toxin receptor expressing mouse brain tumor cells and after orthotopic implantation diphtheria toxin to slow tumor growth. Addition of radiation then further slows growth to increase mouse survival time allowing for the expression of cognitive deficits. However, the effect of radiation alone on mouse survival is not shown. If 48 days post-IR survival (Figure 2) is required for expression of cognitive deficits, why not simply increase the radiation dose? The dose used (3x3.3Gy) is well below that used in the clinic. Moreover, is there an interaction between diphtheria toxin and radiation in terms of tumor response or is it strictly additive? This would be of value when evaluating the addition of a putative radio-sensitizing agent. With respect to new data, the survival response to radiation only (no DT) should be shown.

- Figure 2B is interpreted as showing that radiation and PLX individually prolong survival of tumor bearing mice as compared to DT only, which is apparent from the survival curves and supported by the statistical analyses. However, the combination of PLX and radiation is then claimed to further prolong survival, which is not apparent from the survival curves (the individual curves are similar to the combination treatment) and the statistical analysis was not performed (i.e. radiation only vs. radiation+PLX or PLX only vs. rad+PLX). Moreover, if the effects of radiation only and PLX only on survival are independent, then the combination of radiation and PLX would be expected to result in an additive increase in survival. As presented, it appears that survival after the combination treatment is less than additive, which would suggest that PLX reduced the effectiveness of radiation or that radiation reduced the effects of PLX. The survival experiment shown in Figure 2B is not sufficiently long enough to accurately evaluate the claimed effectiveness of combination treatment and tumor response.

---

## [Author Response]

Essential revisions:1) This DT- Glioma model is useful as it allows for long term evaluation of cognitive deficits and provides proof of principle; however there are several caveats of the GL261 cell line raising questions regarding clinical relevance. Validating key findings using another documented primary tumor model would strengthen the study and claims, however If this is not feasible, it is important to clearly state the caveats and address points raised by reviewer 2 (below) in the Discussion.

We agree with the reviewers on that the current GL261-DTR model alone has some limitations however it represent the first proof of concept study addressing this clinically relevant problem. Given that it is not feasible for us to add another primary tumor model to the current study, we extended our discussion on the caveats of the GL261-DTR model, and articulated on the potential expansion of the application of our proof of concept study. Please also see our response to reviewer #2.

2) The reduction in radiation-induced memory loss is apparent; however data regarding tumor radio response are incomplete and detracts from the strength of this study.Are the effects of radiation alone and PLX alone independent or additive in prolonging survival in tumor bearing mice? Please address specific comments raised by reviewer 3 (below) related to Figure 2B and provide the necessary data and analyses (radiation only vs. radiation + PLX; PLZX only vs. radiation + PLX).

We thank the reviewers for raising this important point and we apologize for not being clear regarding the objectives of the experimental design. The primary objective of our study was to study the effect of CSF-1R inhibition (with PLX5622) on cognitive function in a rodent tumor model that simulated a clinical treatment scenario. From our perspective, a primary limitation of many human-derived tumor models is the requirement for immunodeficient animals. Therefore, we used a rodent glioma cell line (GL261) that is syngenic with C57BL/6J mice which we and others have studied extensively with respect to the cognitive effects of radiation. However, following implantation with GL261 cells, tumors grow rapidly and animals require euthanasia within 3 weeks. Radiation alone in the doses we used does not prolong survival long enough for cognitive assessment. More importantly, patients usually have a therapeutic intervention which reduces tumor burden prior to radiation, which is what we attempted to model with the modified GL261 cell line. Neither diphtheria toxin or PLX5622 were intended to be used as radiation sensitizers or as primary therapeutic tools. To address reviewer 3’s comment, we performed a survival experiment with four groups: tumor only, tumor with radiation, tumor with PLX, tumor with radiation and PLX, and added results and discussion to the revised manuscript (see response to reviewer #3).

3) More in depth and quantitative analyses of microglia numbers using resident and peripheral markers in each of the four conditions in Figure 3. It is difficult to assess from images shown in Figure 3B as mean fluorescent intensity if CD11b is not quantitative or a robust indicator of microglia 'activation'.

We apologize for the lack of clarity on this point we now tried to clarify this and added new data. Figure 3A shows the comparison of CD11b mean fluorescent intensity across groups, this is calculated from flow cytometry analysis. The MFI is a quantitative measure of CD11b expression and has been used as indicator of microglia/macrophage activation (reviewed by Hoogland, et al., 2015). While the original Figure 3B and C were quantification of Iba1 staining covered areas and representative images from immunostaining respectively. In response to the reviewers, to support our claim, we now added quantification of CD68 immunofluorescent staining as Figure 3C. CD68 serves as a lysosomal marker for activated state of microglia/macrophages. We compared areas of Iba1 and CD68 staining across all groups. The new data demonstrated that the CD68 covered area was significantly increased after irradiation, and significantly decreased after PLX treatment. We updated representative images as Figure 3D. In the resubmitted manuscript, we made the following changes to the Results and Discussion.

Updated reference: “Increased CD11b expression level has been used as an indicator of microglia activation during CNS inflammation (Hoogland et al., 2015)”.

Added: “Co-staining for microglia activation marker CD68 suggest that radiation induced a significant increase in activated microglia, which was significantly reduced in animals treated with CSF-1R inhibitor (Figure 3C).”

Added: “Previous reports have demonstrated that cranial irradiation induces morphological changes that correlate with microglia activation, including increased Iba1+ and CD68+ cell numbers and staining coverage (Chiang et al., 1993, Acharya et al., 2016). Taken together with the present results, our results confirmed that cranial irradiation induced microglia activation can be prevented by CSF-1R inhibitor mediated depletion in the presence of brain tumors.”

Materials and methods has been updated to reflect added CD68 staining:

“Samples were sliced into 20 µm coronal sections and stained with rabbit anti-Iba1 (1:100, Wako Pure Chemicals) and rat anti-CD68 (1:200, BioRad) followed by incubation with AF-568 goat anti-rabbit and AF-647 goat anti-rat secondary antibodies (1:500, Life Technologies). […] The area of Iba1+ and CD68+ signals were measured in ImageJ.”

The Figure 3 legend was rewritten to reflect the updated graphs and images.

Reviewer #2:- While the DT-mediated glioma growth control paradigm is novel, simply using a less aggressive – and more molecularly faithful – model of glioma would similarly allow for long term evaluation of cognition after radiation in the context of microglial depletion. My chief concern is the use of GL261 cells – a murine cell line from the 1990s driven by a KRAS mutation (a rare mutation in human gliomas). The molecular characteristics of the tumor can dramatically influence the inflammatory microenvironment of the glioma (see the recent report from the HERBY study, Mackay et al., 2018), and thus the potential influences of the tumor on cognition. Many well-documented xenograft and primary tumor mouse models exist that allow for assessment of cognition at 1 month post-radiation treatment and could be included in this paper to strengthen the claim that microglial depletion can rescue hippocampal-dependent (or hippocampal-independent) cognition following brain irradiation in the context of molecularly definable, faithful glioma models.

We agree with the reviewer’s comment regarding the limitations of the current GL261-DTR model. The specific purpose for developing this tumor model was to mimic what occurs in an actual clinical setting by having the following features: i) an immunocompetent host, ii) fractionated radiation, and iii) reproducible tumor growth allowing evaluation of cognitive function with consistent timing. Although xenograft models that use human glioma cells are effective, they rely upon an immunodeficient host, which will result in a markedly different brain and tumor microenvironment, and different responses to radiation and CSF-1R inhibitor treatment. Unfortunately, adding another primary tumor model to test the cognitive outcomes in a separate cohort of animals is beyond the scope of this particular study, but we recognize that future experiments will need to address this question. Transgenic mice that develop spontaneous tumors rarely do so in a predictable manner and achieving sufficient numbers at similar ages to allow measurement of cognitive function would be a logistic challenge. We recognize that additional model systems will be required to further extend these results.

We added a discussion to clearly state the caveats of the GL261-DTR model, and modified the discussion on the potential expansion of the application of our proof of concept study.

“It is important to acknowledge that the cell line used for the GL261 model is driven by KRAS and p53, a combination of mutation rarely seen in human glioma patients. […] Further studies using genetically modified mouse glioma models that faithfully recapitulate molecular features in glioma patients would help clarify this concern.”

Reviewer #3:- Whereas the reduction in radiation-induced memory loss is apparent, data regarding tumor radio response are incomplete and detracts from the strength of this study.The model uses diphtheria toxin receptor expressing mouse brain tumor cells and after orthotopic implantation diphtheria toxin to slow tumor growth. Addition of radiation then further slows growth to increase mouse survival time allowing for the expression of cognitive deficits. However, the effect of radiation alone on mouse survival is not shown. If 48 days post-IR survival (Figure 2) is required for expression of cognitive deficits, why not simply increase the radiation dose? The dose used (3x3.3Gy) is well below that used in the clinic.

Most of the published rodent studies have used a single dose of radiation (usually 8-10 Gy), which we feel has a very different radiobiologic effect than fractionated irradiation as used for humans. The fractionation scheme we used (3 x 3.3 Gy) reliably produces memory deficits in C57BL/6J strain (Feng et al., 2016), and is comparable to other published reports. Although the treatment paradigm used to treat humans is very different (25-30 fractions of ~2 Gy), individual fractions of radiation in rodents require separate anesthesia each time which may also affect outcomes.

Moreover, is there an interaction between diphtheria toxin and radiation in terms of tumor response or is it strictly additive? This would be of value when evaluating the addition of a putative radio-sensitizing agent. With respect to new data, the survival response to radiation only (no DT) should be shown.

The use of a modified cell line (GL261-DTR) designed to respond to diphtheria toxin was intended to mimic the clinical situation of tumor treatment which results in reduced tumor burden prior to radiation treatment. To further address this point, we added the following treatment groups, tumor only, tumor + PLX, tumor + fWBI and tumor + PLX + fWBI, N=5 each group. The new data shows that radiation treatment alone significantly prolonged survival compared to the tumor only group (30 vs. 20 days of median survival, p<0.01), but this was not long enough for us to conduct behavior assay on the animals. In the survival dataset from our original submission, DT treatment alone was able to extend median survival to 39 days, and that DT + fWBI has an undefined median survival, These results suggest that there is a synergistic effect of combined DT and fWBI in tumor response. Therefore, DT in our treatment paradigm acted as a radiosensitizer, although it was not intended to be used as one.

On the other hand, added PLX to radiation treatment prolonged survival compared to radiation alone (40 vs. 30 days, p = 0.087) group. This survival advantage (20 days longer than tumor alone) was longer than each single treatment group (4 and 10 days for PLX and fWBI groups, respectively). Therefore, tumor response to PLX + fWBI was more than additive than each single treatment. However, survival curves of fWBI and PLX + fWBI are not significantly different.

Given that treatment interaction is not the main focus of the current study, we added these new data and analyses as supplemental figures (updated in Figure 2—figure supplement 1), and added the following text to the manuscript.

“Next, we tested possible treatment interactions using single and combined treatments of DT, CSF-1R inhibitor and fWBI. […] In light of these results, we tested the effects of CSF-1R inhibition on cognition in GL261-DTR bearing mice after DT and fWBI treatments.”

Original text: “We found that fWBI or CSF-1R inhibitor treatment alone significantly improved survival …”

Changed to: “We found that DT + fWBI or DT + CSF-1R inhibitor treatment significantly improved survival …”

Original text: “Combined CSF-1R inhibitor and fWBI treatment …”

Changed to: “Combined DT, CSF-1R inhibitor and fWBI treatment …”

- Figure 2B is interpreted as showing that radiation and PLX individually prolong survival of tumor bearing mice as compared to DT only, which is apparent from the survival curves and supported by the statistical analyses. However, the combination of PLX and radiation is then claimed to further prolong survival, which is not apparent from the survival curves (the individual curves are similar to the combination treatment) and the statistical analysis was not performed (i.e. radiation only vs. radiation+PLX or PLX only vs. rad+PLX). Moreover, if the effects of radiation only and PLX only on survival are independent, then the combination of radiation and PLX would be expected to result in an additive increase in survival. As presented, it appears that survival after the combination treatment is less than additive, which would suggest that PLX reduced the effectiveness of radiation or that radiation reduced the effects of PLX. The survival experiment shown in Figure 2B is not sufficiently long enough to accurately evaluate the claimed effectiveness of combination treatment and tumor response.

The survival curve of the PLX and radiation combined treatment is not significantly different from the PLX or radiation treatment alone. The end point of this experiment was originally set to day 60 after tumor inoculation (mistakenly shown as 55 days in the original submission). We agree that the time was not long enough to assess our original claim. We have corrected the wording in the Results.

Original text: “Combined CSF-1R inhibitor and fWBI treatment further improved survival of tumor bearing mice (Figure 2B, black line).”

Changed to: “Combined DT, CSF-1R inhibitor and fWBI treatment did not improve the survival time of tumor bearing mice (Figure 2B, black line).”

We apologize for the error in the length of survival curve in the original submission. It has been corrected in updated figure 2, and Figure 2—figure supplement 1A and B. The statistics in our original submission were obtained using the day 60 data, there was no change regarding this matter.

In addition to these changes, we added a Key resource table at the start of the Materials and methods.